# Distance of flight of cosmic-ray muons to study dynamics of the upper muosphere

Hiroyuki K.M. Tanaka[1,2]*

1. University of Tokyo, Tokyo, Japan.
2. International Virtual Muography Institute (VMI), Global, Tokyo, Japan.

Correspondence to: Hiroyuki K.M. Tanaka

*email: ht@eri.u-tokyo.ac.jp

**Abstract**

The Earth can be divided by main layers, including the atmosphere, geosphere (solid Earth), and biosphere, depending on its predominant component. In this work, the layer of the Earth which constantly contains a high concentration of muons ($\sim 8 \times 10^{12}$ muons) and its upper border are respectively defined as the muosphere and muopause. The altitude of the muosphere spans from the lower stratosphere to the upper crust of the Earth. In order to study its dynamics, the muopause height was spatiotemporally studied with a new kind of technique called the distance of flight (DOF) which utilizes variations in the muon's decay length. In this work, (A) numerical modeling was performed, and it was clarified that seasonal variations in the cosmic muon flux are predominantly ruled by muopause dynamics, (B) the muon data were compared with the balloon-based measurement results, and it was confirmed that muopause dynamics is closely related with lower-stratospheric height variations. Since the muopause is the region spanning between the upper troposphere and the lower stratosphere, the potential of the current distant of flight (DOF) approach needs to be further investigated by cross-comparing related case studies and other atmospheric climate datasets.

**Introduction**

Muons are secondary particles generated in the Earth's atmosphere as a result of hadronic interactions between the incident primary cosmic rays (primaries) and atmospheric nuclei such as nitrogen and oxygen. These primaries rarely interact with matter within the top

region of the Earth's atmosphere due to the low number density of atmospheric nuclei. However, as primaries' injection depth increases, the density of the atmosphere increases, and these primaries increasingly interact with nuclei, producing mesons such as charged pions and kaons which eventually decay into muons. Once these mesons decay into muons, there will generally be no further interaction to generate new particles since muons do not strongly interact with matter since muons undergo only electromagnetic and weak interactions but not strong nuclear interactions. Therefore, muons are extensively produced within particular altitude regions of the atmosphere. The muons' production rate increases as a function of the atmospheric depth such that: <0.01%, ~0.2%, ~2%, ~30%, and ~80% (of the entire muons we observe at sea level) up to an atmospheric depth of 5 hPa, 10 hPa, 50 hPa, 100 hPa, and 200 hPa, respectively, and almost all muons are generated up to an atmospheric depth of 300 hPa (Particle Data Group, 2022; Boezio *et al.*, 1999; Boezio *et al.*, 2000). On the other hand, due to their strong penetration power, the muons also exist in the geosphere with a rock/water depth up to ~5/~10 km. Consequently, muons are predominant in the region defined on the altitude coordinate as ranging from 30 km and -10 km which also partly overlaps with regions of the humanosphere. This region (from +30 km to -10 km from sea level) is here defined as the muosphere. Accordingly, the muopause is defined as the upper boundary of the muosphere (as with the tropopause defining the upper boundary of the troposphere) which is located at 30 km asl. The key characteristics of the muons within the muosphere are: (A) an abundance of ~$8.0 \times 10^{12}$ muons[*] with the number of ~$5 \times 10^3$ muons km$^{-3}$ are constantly present in the muosphere, (B) ~$8 \times 10^{16}$ muons are generated in the muosphere every second and (C) ~$5 \times 10^{16}$ muons arrive at sea level every second. The exception to this would be neutrino-induced muons which exist throughout the geosphere (Particle Data Group, 2022), but the number of these neutrino-induced particles within the geosphere is too small (< $10^{-9}$ muons km$^{-3}$) to categorize them as being part of the muosphere.

[*]$(1.6 \times 10^2 \, \mathrm{m^{-2} s^{-1}})$ [averaged muon flux] $\times$ ($5 \times 10^{14} \, \mathrm{m^2}$) [Earth's surface area] $\times$ ($3.0 \times 10^4$ m) [thickness of the muosphere] / ($3 \times 10^8 \, \mathrm{m s^{-1}}$) [speed of muons]

The thickness of the muosphere spatiotemporally fluctuates due to processes near the surface of the Earth: mainly crustal deformation and land temperature variations. Crustal deformation alters the density of the shallow crust and local topography. When this occurs, the underground depth threshold for muons to reach will be altered; hence the position of the bottom part of the muosphere is regionally altered, but the time scale of this change

is very long (over millennia). On the other hand, the near-surface temperature variations
will alter the isobaric surface height near the muopause in much shorter time scales.

Since muopause height variations are closely related with the upper-tropospheric and
lower-stratospheric isobaric surface height variations, studying muopause dynamics has
the potential to contribute to research in this field. For example, it was reported that the
2020 and 2021 ozone holes were both associated with large decreases in polar lower
stratospheric heights (Yook *et al*., 2022). Sudden stratospheric warming (SSW) is
characterized by large isobaric surface height rises at the pole (Kretschmer *et al*., 2018).
The 2022 Hunga Tonga–Hunga Ha'apai volcano eruption, Tonga resulted in a substantial
injection of water vapor into the upper atmosphere (Millán *et al*., 2022; Vömel *et al*.,
2022). Such changes in the atmospheric composition should have had a noticeable impact
on the muopause.

The established muographic imagery techniques have been applied to natural phenomena
such as volcanoes (Tanaka *et al*., 2014), cultural heritage (Morishima *et al*., 2017), tropic
cyclones (Tanaka *et al*., 2022a), meteotsunami (Tanaka *et al*., 2022b), and contraband
detection (Gnanvo *et al*. 2011). These techniques take advantage of known properties of
muon transmission and scattering through matter. In this work, the DOF technique is
added, and it will be shown that the muopause height variations can be measured with
this technique based on the quantitative analysis of the time-sequential muon data. Since
muons are leptons with a lifetime at rest is 2.2 microseconds, the actual lifetime as
observed from the detector, and therefore the path traveled, is much longer by the
relativistic time dilation factor, and that the latter depends on energy. Consequently, the
sea-level muon flux will decrease/increase as the muopause uplifts/lowers. This is the
basic principle of the DOF approach. In this work, the aim was to show balloon-based
lower stratospheric height variations are well reproduced by applying the DOF approach
to the time-sequential muon observation data.

There are a number of reports exploring barometric and temperature effects in the muon
flux (Tanaka *et al*., 2022a; Tilav *et al*. 2010; COSINE-100 Collaboration, 2020;
Myssowsky and Tuwim *et al*. 1926; Blackett, 1938; The IceCube Collaboration, 2019;
Adamson *et al*., 2010; Tramontini *et al*., 2019; Blumer *et al*., 2005; Dmitrieva *et al*. 2011).
However, many of these works focus on either tropospheric barometric effect or
stratospheric temperature effect. In this work, DOF approach was modeled and applied
to the 1,044-day time sequential muon data to compare with the Japan Meteorological
Agency's balloon data.

As a result, the following two major characteristics were identified: (A) seasonal
variations in the muon flux due to the isobaric surface height effect are much larger than
seasonal variations due to the barometric effect, therefore, (B) the isobaric surface height
derived by the DOF technique is consistent with the balloon-based upper-atmosphere
isobaric surface height measurement results. In this paper, a detailed description of the
process to arrive these results is provided. A brief discussion of its current limitation and
potential improvements are also described.

**2. Muosphere and muopause**
If the *abundant muon flux* is defined as $\geq$10 muons $m^{-2}s^{-1}sr^{-1}$, the abundant muon flux is
available within the altitude region between 24.5 km above sea level (a.s.l) to 20 m below
ground surface of 40 m below sea level (b.s.l). This space is hereafter defined as the
"middle muosphere". The region where muons exist above the middle muosphere is
called the upper muosphere, and the region where muons exist below the middle
muosphere is called lower muosphere. The muon flux available in the middle muosphere
and the lower muosphere on the Earth are summarized in Table 1. Vertical classification
of the muosphere is shown in Figure 1 as a function of the available muon flux. The
muopause is defined as the altitude region with the highest muon generation rate per unit
atmospheric depth ($dI/dX$, where $I$ is the muon flux, and $X$ is the atmospheric depth).
$dI/dX$ takes maximum is ~21.5 km a.s.l. where a muon flux increases by ~40 $m^2s^{-1}sr^{-1}$
every 1 km atmospheric depth. (Figure 2). This peak is defined as the *specific muopause*.
If the width of the muopause is defined the region where $dI/dX \geq 20$ $m^{-2}s^{-1}sr^{-1}km^{-1}$ (half
maximum), the muopause spans from 18 km a.s.l. to 24.5 km a.s.l. This muopause is
defined as the *full-width half-maximum (FWHM) muopause*. The middle muosphere
contains both atmospheric and geospheric layers.










Table 1. Available muon flux in the middle muosphere and the lower muosphere on the
Earth. The data were taken from Particle Data Group (2022).

| | Height (km) | Muon flux ($m^{-2}$ $sr^{-1}$ $s^{-1}$) | |
|---|---|---|---|
| Upper Muosphere | 30 | $<10^{-2}$ | |
| - | | | |
| Middle Muosphere | 25 | $1.0 \times 10^{1}$ | Atmosphere |
| | 23 | $3.0 \times 10^{1}$ | |
| | 20 | $1.5 \times 10^{2}$ | |
| | 15 | $2.2 \times 10^{2}$ | |
| | 10 | $2.3 \times 10^{2}$ | |
| | 5.0 | $2.0 \times 10^{2}$ | |
| | 3.0 | $1.5 \times 10^{2}$ | |
| | 2.0 | $1.2 \times 10^{2}$ | |
| | 1.0 | $1.0 \times 10^{2}$ | |
| | Sea level | $9.0 \times 10^{1}$ | - |
| | -0.02 | $1.0 \times 10^{1}$ | |
| - | | | |
| Lower Muosphere | -0.05 | $2.8 \times 10^{0}$ | Geosphere |
| | -0.01 | $3.2 \times 10^{-1}$ | |
| | -0.5 | $1.2 \times 10^{-2}$ | |
| | -1.0 | $1.2 \times 10^{-3}$ | |
| | -2.0 | $7.0 \times 10^{-5}$ | |
| | -3.0 | $6.0 \times 10^{-6}$ | |
| | -4.0 | $6.0 \times 10^{-7}$ | |
| | -5.0 | $7.0 \times 10^{-8}$ | |

174

175

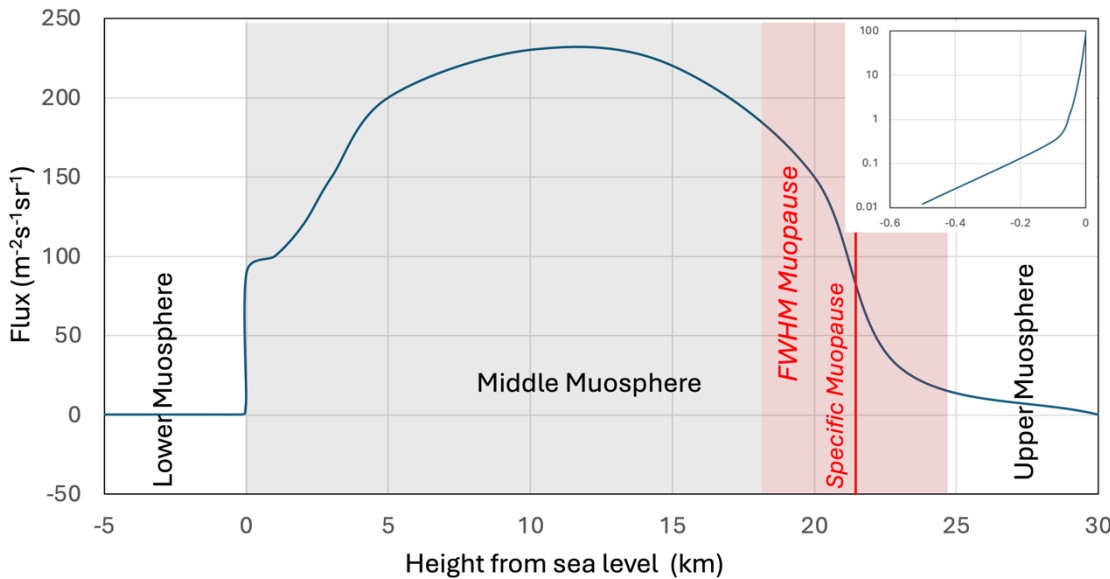

Figure 1. Configuration of the muosphere. The gray and red shaded areas respectively indicate the middle muosphere, and the FWHM muopause. The vertical red line indicates the location of the specific muosphere. The inset indicates a magnified view in the vicinity of the interface between the middle muosphere and the lower muosphere.

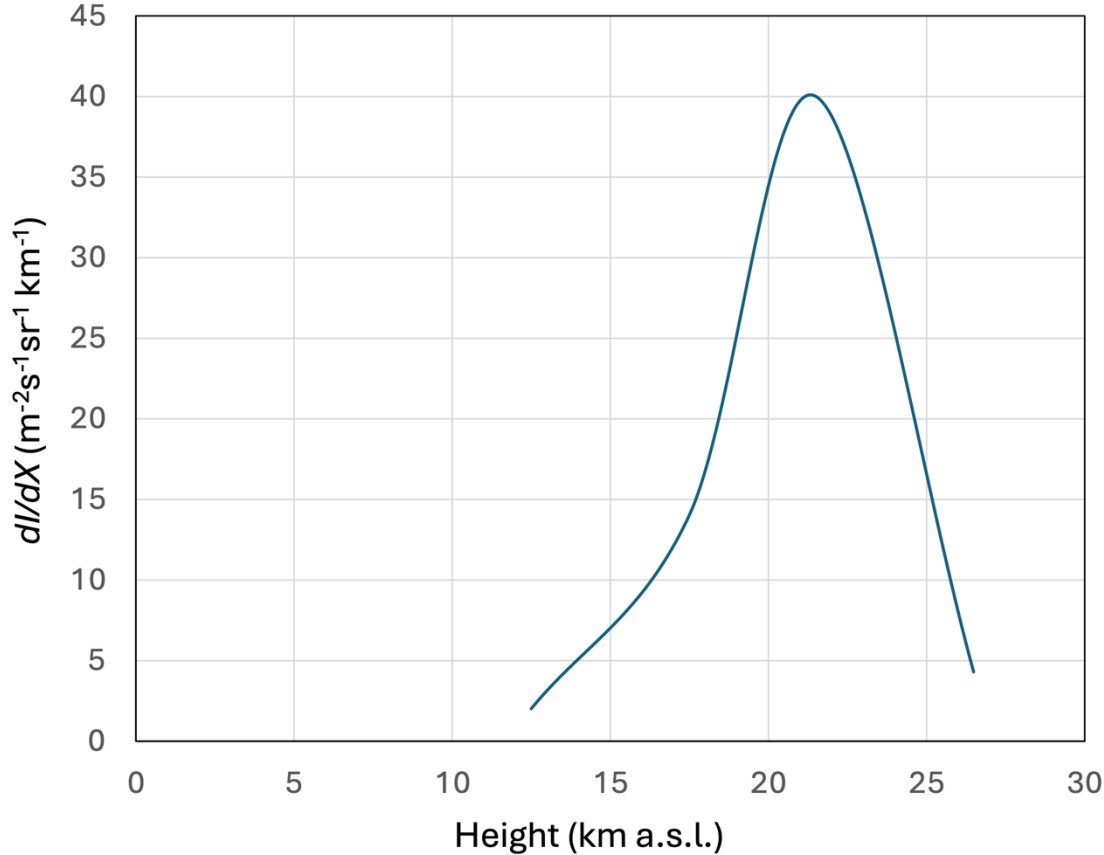


Figure 2. Muon flux gradient as a function of the altitude.

The proposed DOF technique is similar to ground-based atmospheric LiDAR in terms of
scanning the upper atmosphere. Muons are generated most extensively at the muopause
that spans from the upper troposphere to the lower stratosphere. If the tropopause shifts
upward, this density gradient shifts upward accordingly, and accordingly, the muopause
shifts upward, then the muon's travel distance increases. The current technique measures
the height of where muons are extensively generated; hence the height of the tropopause
by measuring the muon's DOF. For this aspect, the DOF technique measures the dynamic
(PV) tropopause height variations indirectly. (UV LiDAR measures the $O_3$ tropopause,
and visible light LiDAR measures the cirrus cloud.)  Due to the muon's strong penetrating
capability, the DOF technique is not influenced by the cloud existence.


**2. Principle of the DOF technique**
The atmospheric cascades of secondary pions and kaons are developed as a result of the
competition between hadronic collisions of mesons with nuclei and the decay process of
the mesons in the atmosphere. Therefore, muons are not generated at a specific altitude,
but instead they are generated within a certain altitude range (Boezio *et al.*, 1999; Boezio
*et al.*, 2000).
Figure 3 shows the layer span of the muosphere on the Earth and the principle of the DOF
technique. As shown in Figure 3A, the muosphere covers the region from the lower
stratosphere, troposphere, and shallow region of the geosphere (shallow crust and ocean).
Topography of the muopause is determined by the isobaric surface height distribution of
the upper atmosphere, and is generally related to the height of tropopause. However, the
tropopause region does not usually overlap with the muopause region. The isobaric
surface height is high when the surface temperature is high and low in when the surface
temperature is low since the larger vertical temperature gradient causes deeper convection
in the troposphere, pushing the isobaric surface, upwards; hence seasonally varied. More
detailed descriptions can be found later in the "Balloon-based studies near the muopause"
section. As shown in Figure 3B, variations in the height of the muopause will affect the
muon generation point; hence the muon's DOF. The number of muons decreases when
the muopause is uplifted. If the isobaric surface height effect is comparable to or stronger
than the barometric effect on the muon flux, the spatiotemporal variations in the
muopause can be measured by using the local barometric data. Detailed descriptions
about two essential aspects of the DOF technique, (1) modelling of the seasonal
barometric effect on the muon flux and (2) modelling of the seasonal isobaric surface
height effect on the muon flux, are given in the following subsections.
There are three effects that influence the muon flux: (A) variations in the muon's range
due to barometric variations, (B) variations in the muopause height due to variations in
mass exchange between the upper troposphere and the lower stratosphere; hence
variations in the muon's distance of flight between the muopause and sea level, and (C)
variations in hadronic interaction mean free paths due to stratospheric temperature
variations. There are three effects in total. Effect (C) comes from the competition between
the pion's and kaon's hadronic interaction mean free path (MFP) and decay length. If the
stratospheric temperature increases, the air thermally expands and thus, the hadronic
interaction MFP increases. However, this effect is much smaller than the other two effects
since this factor is only relevant for muon flux in the energy region above 50 GeV
(IceCube Collaboration, 2019) where the integrated muon intensity is lower by more than
2 orders of magnitude. Effect (A) includes variations in the muon's stopping power due
to variations in their energy loss rate in the atmosphere, and variations in the muon's

decay length due to variations in their deceleration rate in the atmosphere. While there have been several works investigating the Earth's atmosphere using muons, using effect (A) (Tanaka *et al*., 2022b; Tramontini *et al*. 2019), these previous works focused on gauging areal density variations (convertible to temperature and pressure) of the atmosphere which is conceptually close to muographic imagery. On the other hand, the DOF technique inversely using the effect (B) has never been attempted.

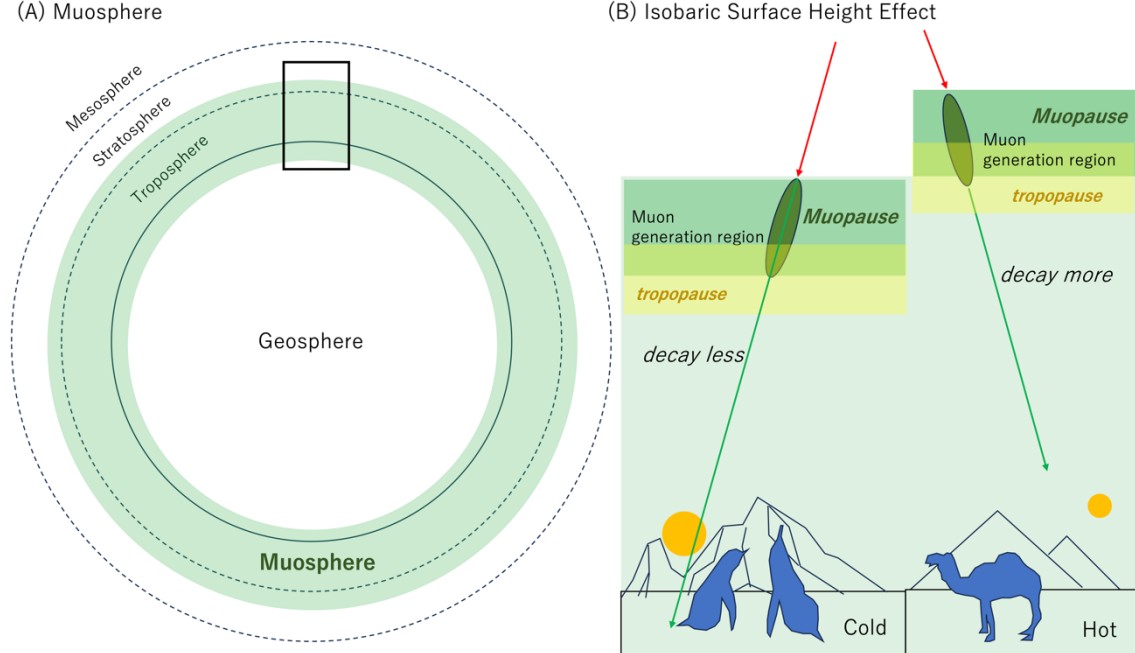

Figure 3. Definition of muosphere and the principle of the DOF technique. The span of the muosphere is shown along with other layers of the Earth (A). The black box indicates the muosphere for the region shown in (B). Red arrows and green arrows respectively indicate the primary cosmic rays and muons. Additionally, (B) shows an example of the contrast between the average height of the muopause above colder surface temperature and the height of the muopause above warner surface temperature. Dark green ovals indicate the muon production regions. As is indicated with yellow-filled boxes, the tropopause and the muopause do not exactly overlap with each other.

## 3. DOF Modeling

### 3.1 Modeling of the seasonal barometric effect on muon flux

The cosmic muon flux is also influenced by ground-level barometric variations because the amount of muon energy loss depends on the total areal density along their trajectories.

In this work, we took advantage of muon flux variations associated with the presence of
a cyclone to derive variations in the muon counts in the detector as a function of the
ground-level pressure in Kagoshima city. The advantage of using the cyclone data is that
since the cyclone moves quickly (typically within 24 hours) and will dramatically alter
the ground-level atmospheric pressure (sometimes by more than 40 hPa), barometric
muon flux variations can be evaluated without being influenced from the longer-time-
scale isobaric surface height effect. According to Tanaka *et al*. (2022a), the barometric
correction of the muon flux can be reasonably performed by using the tropic cyclone
passage events. Their reported flux drop rate is 0.0016/hPa (theory) and 0.001/hPa -
0.002/hPa (observation). This flux drop rate includes (A) the flux drop due to the higher
rate of muon's absorption into the atmosphere (stop and decay), and the flux drop due to
an increase in inflight decay of muons (since they lose their energy more). Figure 4A
compares the temporal variations in the muon flux and the temporal variations in the
ground-level atmospheric pressure induced by the 2018 Typhoon No 24. Figure 4B shows
the relationship between the muon flux and the atmospheric pressure (both measured in
Kagoshima city). The metrological data in Figure 4A were taken from Reference (Japan
Metrological Agency, 2023). The data points in Figure 4B were fitted by a linear function
and the result is shown in the following equation:

$\Delta N = - 0.0012 \Delta P$ [hPa] + 2.2159.                            (1)

This result indicates that the muon flux varies by 1.2% if the ground-level atmospheric
pressure changes by $\Delta P = 10$ hPa (with respect to 1,000 hPa). Since the detector used for
measuring this cyclone effect on the muon flux is identical to that used for the DOF
measurements, the external factors including the zenith angular dependence of the muon
flux, geometrical acceptance of the detector, etc. are canceled out. The fractions of the
number of the data points are respectively 60%, 82%, and 100% for deviations of ≤1σ,
≤1.5σ, and ≤2σ from the estimated line. The SD of the data points from the estimated line
($6.5 \times 10^{-4}$) is close to the statistical error associated with the data points ($5.2 \times 10^{-4}$ - 5.4
$\times 10^{-4}$). The $R^2$ value (coefficient of determination) for this fitting was 0.85. The current
result (0.00055/hPa -0.00185/hPa) in agreement with the flux drop rate reported in the
prior work (0.001/hPa -0.002/hPa) within the error bars. The difference between them
(0.0037) can be the fitting uncertainty which adds an uncertainty of ~18 m in estimation
of the muopause height (See below). Eq. (1) was used for the barometric correction to the
muon flux in the current work.

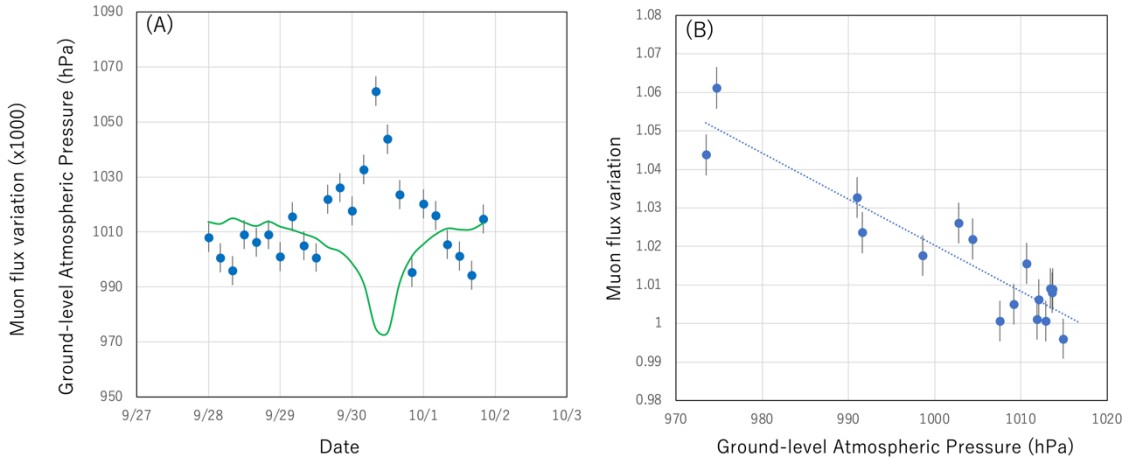


Figure 4. Variations in the muon flux induced by the ground-level atmospheric pressure
variations. The muon flux variations (blue filled circles) are compared with the ground-
level atmospheric pressure variations (green solid lines) induced by the 2018 Typhoon
No 24 (A). The muon flux variations (blue filled circles) are shown as a function of the
ground-level atmospheric pressure. The numbers on the vertical axis indicate the muon
flux variation times 1,000. The dotted line indicates the linear function fitted to these data
points (B).


With Eq. (1), seasonal variations of the muon flux caused by variations in the ground-
level atmospheric pressure (P-driven muon flux variations) were evaluated. Figure 5A
shows seasonal variations in the ground-level atmospheric pressure measured at the
Kagoshima Meteorological Observatory in the period between August 2017 and August
2020. Figure 5B shows the corresponding P-driven muon flux variations based on Eq. (1).

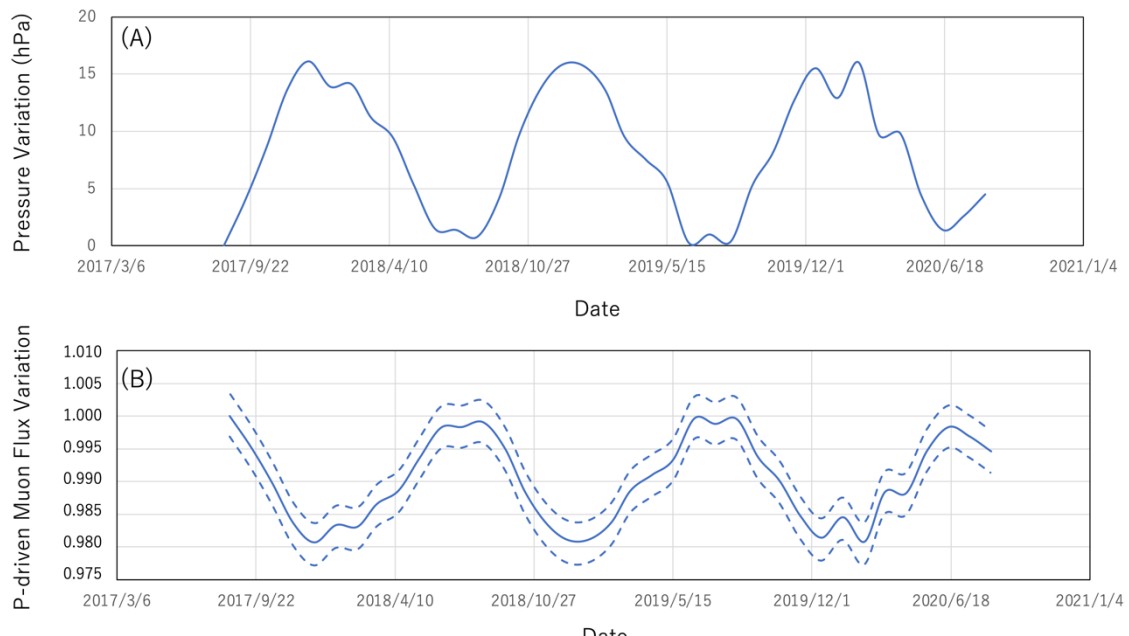

Figure 5. Seasonal variations in the ground-level atmospheric pressure (A) and estimated variations of muon flux due to variations in the ground-level atmospheric pressure (B) based on Eq. (1). The data are shown in the period between August 2017 and August 2020. The ground-level atmospheric pressure data were taken from Reference (Japan Metrological Agency, 2023). The area between dashed lines in panel (B) indicates the barometric modeling error (1 S.D.) The error values associated with the barometric data are not published by Japan Metrological Agency.

## 3.2 Balloon-based studies of the isobaric surface height near the muopause

Japan Meteorological Agency launches a balloon from Kagoshima city twice a day (09:00 and 21:00 JST) to monitor the isobaric surface height of the upper atmosphere. The monthly isobaric surface height measurement results acquired in 2018 and 2019 are shown in Figure 6. As shown in this figure, the altitude of the muopause varies $\Delta H \sim 500$ m, reflecting seasonal variations (i.e., altitude that increases in the summer time) of the muopause. While the muon generation depth has a certain span (50-300 hPa), as can also be seen in this figure, the isobaric surface height corresponds closely with the variations in this span. Therefore, we can conclude that the isobaric structure of the upper muosphere is simply pushed further from sea level in summer and pushed closer to sea level in winter (dark green ovals in Figure 3B). Consequently, it is expected that variations in the muon survival rate at sea level is a function of the muopause height.

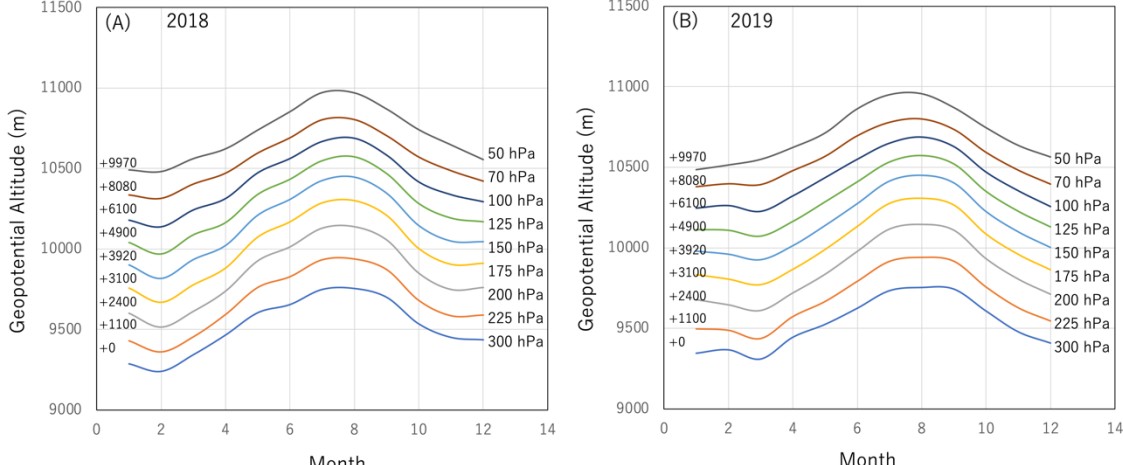

Figure 6. Seasonal changes in the isobaric surface height of the upper troposphere and the lower stratosphere. The data acquired in 2018 (A) and the data acquired in 2019 (B) are shown. The data were taken from the Japan Metrological Agency survey (Japan Metrological Agency, 2023). The numbers on the left side of each panel indicate the offset of the altitude in units of meters.

**3.3 Modeling of the seasonal isobaric surface height effect on the muon flux**

As was described in the previous subsection, the isobaric surface height variations $\Delta H$ ($H$) are independent from $H$ in this span. The modeling procedure is summarized as follows.

(A) The zenith-angular dependent open-sky muon spectrum data points are taken from various prior experimental works (Allkofer *et al*., 1985; L3 Collaboration., 2004).

(B) These muon spectrum data points are interpolated to derive $I_0(E,\theta)$ by using the Thompson and Whalley analytical formula (Thompson and Whalley, 1977).

(C) Calculations of the angular-dependent muon flux are done, based on the following formula:

$$n(\theta,\phi) = \int I(E,\theta)dE, \qquad\qquad (2-1)$$

$$I(E,\theta) = I_0(E,\theta) \exp[-\Delta H(\sin\theta)^{-1}(c\tau\gamma)^{-1}](1-\Delta N), \qquad (2-2)$$

where $E$ and $\theta$ are respectively the muon's energy, $\tau$ is the muon's decay constant, and the arrival angle from zenith at sea level, $I_0(E,\theta)$ is the reference muon flux, and $\gamma$ is the Lorentz factor. In Eq. (2.2), reduction of $E$ is calculated by assuming the fixed atmospheric pressure (1013 hPa), and the effect of temporal barometric variations on $dE/dX$ was neglected since the decay effect coming from the muon's energy loss variations due to temporal barometric variations is already incorporated into $\Delta N$. Moreover, this decay effect in seasonal variations is small. For example, there is 15 hPa barometric variations between winter and summer (Figure 3), and these variations induce variations in energy loss of 30 MeV (in vertical), which extend/contract the muon's decay length by 190 m (in vertical) which is smaller than the muopause height variations by a factor of 3 (Figure 6). Radiative processes (i.e., bremsstrahlung, direct pair production, and photonuclear interactions) in $dE/dX$ were neglected due to relatively long radiation length in air. Figure 7 plots Eq. (2-1) for $\Delta H = 0$ m (Figure 7A) and $\Delta H = +500$ m (Figure 7B). The data points are overlaid on this plot. Although it is difficult to estimate $\Delta H$ when these data were gathered, but it is assumed that $\Delta H = 0$. It is important to note that in the current work, the quantity to be evaluated is variations in $H$, not $H$ itself. The positive and negative signs attributed to $\Delta H$ respectively indicate respectively upward variations and downward variations. If $\theta$ approaches the value of 90°, $(\cos\theta)^{-1}$ will be diverged, so in this case, the spherical curvature of the Earth has to be considered (for $\theta$ ~90°).

(D) Calculate the number of muons recorded by the detector with:

$$N = \int_{\phi_0}^{\phi_1} \int_{\theta_0}^{\theta_1} n(\theta,\phi)d\theta d\phi, \qquad (3)$$

where $\theta_0$, $\theta_1$, $\phi_0$, and $\phi_1$ are respectively the detector's zenith ($\theta_0$-$\theta_1$) and azimuth ($\phi_0$-$\phi_1$) angular acceptance. Eq. (3) was used for the isobaric correction to the muon flux in the current work.

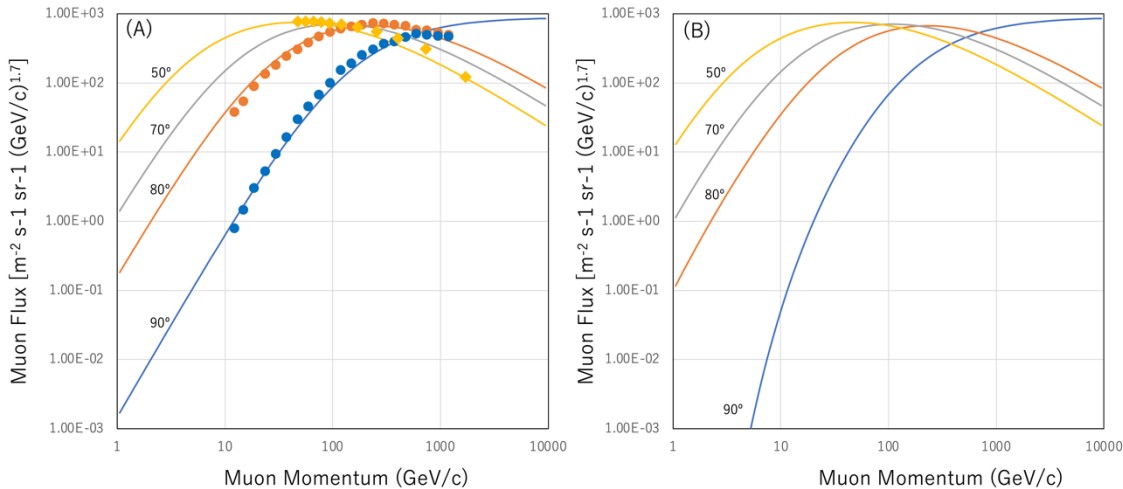


Figure 7. Differential muon flux for different isobaric surface altitudes. The spectra
calculated for the reference isobaric surface altitude ($\Delta H = 0$ m) are shown in (A) and for
the case when the isobaric surface is uplifted by 500 m in (B) for various muon's arriving
angles: 90° (blue), 80° (orange), 70° (gray), and 50° (yellow). Only slanted muons are
shown due to the geometrical configuration of the current detector setup (see below).
Filled circles and filled rhombuses are respectively the data points taken from Allkofer *et*
*al*. (1985) and L3 Collaboration (2004).


With Eq. (2-1), seasonal variations of the muon flux due to the isobaric surface height
effect (H-driven muon flux variations) were calculated. Figure 8A shows the balloon-
based $\Delta H$ value averaged over the altitudes which ranged between 50 hPa and 300 hPa.
Figure 8B shows the corresponding H-driven muon flux variations based on Eq. (2-1). In
order to match the angular acceptance of the tracker (described in the next section), the
zenith-angular integration range of Eq. (2-1) was set to be 50°-90°. It was assumed that
the muon's arriving angles are azimuthally isotropic. These results are subsequently used
for the muon flux modeling process which will be described in the following subsection.
As can be compared between Figure 5 and Figure 8, the seasonal isobaric surface height
effect (up to 8%) is much larger (by a factor of 4) than the seasonal barometric effect (up
to 2%).

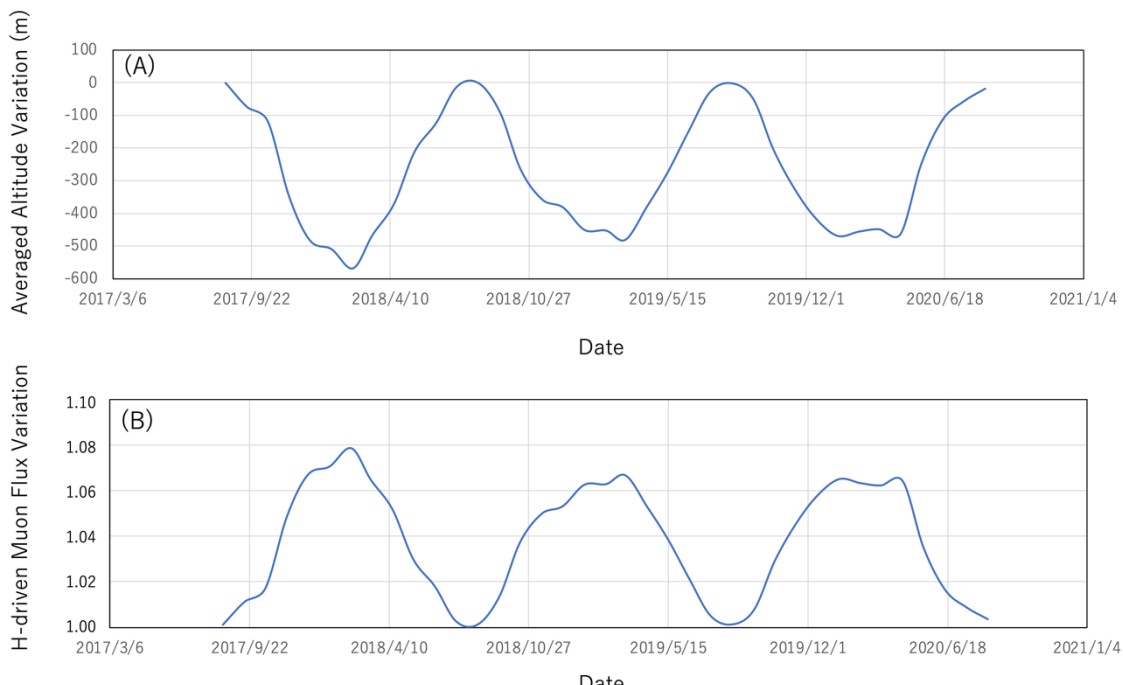

Figure 8. Balloon-based isobaric height variations averaged over the altitudes between 50
hPa and 300 hPa (A) and H-driven variations of the muon flux without the barometric
correction (B). The data are shown for the period between August 2017 and August 2020.
The balloon-based isobaric height data were taken from Japan Metrological Agency
(2023) The error values associated with the balloon data are not published by Japan
Metrological Agency.

**4. Apparatus**
The muon tracker used in this study consisted of 90 scintillator strip detectors. Each
scintillator strip detector consisted of a plastic scintillator (Eljen EJ-200) strip connected
to a photomultiplier tube (PMT; Hamamatsu R7724) via an acrylic light guide. The
typical pulse height outputted from the PMTs were 3-5 V while the threshold level of the
discriminator was set to be 50 mV, so that the counting rate would not be easily influenced
by the drift of the PMT gain and the discriminator's threshold level caused by variations
in ambient temperature. The width and the length of the plastic scintillator strip were
respectively 100 mm and 1500 mm. These strips were arranged vertically and horizontally
to form three position sensitive detectors (PSDs). Three PSDs were vertically arranged
with a spacing of 60 cm. In order to reject electromagnetic components, such as
positrons/electrons a 10-cm thick lead slab concealed inside a 1.5 cm thick stainless-steel
case (totaling 3 cm in thickness) were inserted into each interval between the PSDs. A
stainless-steel case is needed to protect the lead for the following reasons: (A) lead is soft,
and for long-time measurements, it can be deformed, so it has to be supported by the
stainless-steel case, and (B) lead is poisonous, not allowed to use it outdoor environments
without coverage, so it has to be covered by the stainless-steel box.

The best way to check the long-term detector stability is to use IBE (inverse barometric
effect). Barometric variations in the muon flux are compensated by the tidal height
variations since the total areal density above the detector will be constant as long as the
detector is located undersea, so that the local fluctuations due to barometric variations can
be intrinsically removed from the data without any artificial actions. According to the
data taken at the Trance Tokyo-bay Aqua Line undersea tunnel, Japan, variations in the
lunar daily muon rate was ~0.0028 (S.D.) including statical errors of ~0.001 (S.D.) for
half-year measurements (Tanaka *et al*. 2021). The detector had the same configuration
used in this work (R7724 and EJ200).

Each of the resultant PSDs consists of a segmented plane with $15 \times 15$ segments having
a $1.5 \times 1.5 \ m^2$ active area with a spatial resolution of 10 cm. Since the distance between
the uppermost stream detector and the lowermost stream detector is 120 cm, the angular
resolution of this detector is 83 mrad. This angular resolution is equivalent to the spatial
resolution of 830 m at a location 10 km from the tracker, but it is reduced to 8.3 km at a
location 100 km from the tracker. The elevation and azimuth angular acceptance are
respectively 0 ° - 51° and ±51°. However, since the active area is drastically reduced for
muons injecting at higher angles with respect to PSD planes (e.g., for muons arriving at
an elevation angle of 51° and an azimuth angle of 51°, the tracker's active area is reduced
to 1/225), for practicality, a much smaller angular region (14 °- 32 ° for elevation angular
region and ±28° for azimuthal angular region) was employed. For tracking, all vertices
are examined but only the vertices that are aligned (along a straight line) are counted as
an event to ensure that only muons were selected. Lead and stainless-steel shields within
the detector decrease the background noise, however they also increase the possibility of
muon scattering events. However, these scattering angles (10-20 mrad) are considered to
be negligible in comparison to the current tracker's angular resolution (>80 mrad). The
muon tracker used in the current experiment was located in Kagoshima city, Japan and it
was pointed towards the southern direction. The measurement period was between
August 20, 2017- June 30, 2020 (1,044 days).

## 5. Comparison between the model and the experimental data

Figure 9 shows the seasonal variation in the muon flux data acquired in the period between August 20, 2017- June 30, 2020. As was expected, the muon flux showed a negative correlation between the ambient temperature and the muon flux, indicating that the isobaric surface height effect is predominant in seasonal variations in the muon flux. In this period there were not any specific extraterrestrial events (such as a Forbush decrease) that could have affected the primary flux. The muon counts in each bin (bin width = 3 days) ranged between $7\times10^5$-$7.4\times10^5$. The muon flux variations were normalized to the value observed on August 20, 2017. The muon flux modeling results with barometric and isobaric corrections are overlaid on this plot (red solid lines in Figure 9). The root mean square (RMS) of the deviations between the theoretical values and the observational values is 0.987%. The measured seasonal variation in the muon flux is well explained by combining the current barometric and isobaric correction models.

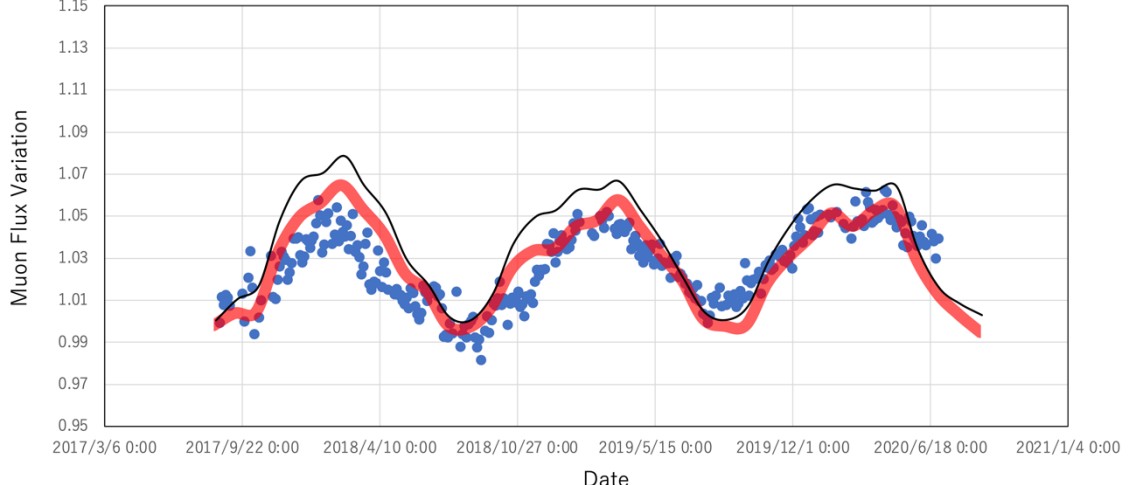

Figure 9. Seasonal variation in the muon flux data acquired in the period between August 20, 2017- June 30, 2020. The observation values (blue filled circles) and theoretical values (red solid lines) are shown. The statistical errors associated with each data point fit within the size of the circles. The barometric modelling errors are within the width of the red solid lines. Black solid lines indicate the predicted flux without the barometric correction.

## 6. Limitations and potential improvements

The RMS deviation of the observed muon flux variations from the DOF modeling results

(~1%) induces an error of ~60 m in the estimation of the muopause height with the DOF
technique. The DOF time resolution is 3 days. These characteristics, the accuracy and
time resolution, are both significantly lower than the accuracy and the time resolution that
can be attained with the GPS-loaded balloons (~5 m and 1 second). This is the main
limitations of the DOF approach in its current stage of development. These limitations
mainly come from (A) the statistics and (B) the modeling accuracy. Regarding factor (A),
the detector size needs to be enlarged to record more muon events. In order to confirm
this detector size effect, as the first step, the detector size will be increased to a double
size (4.5 m$^2$) to verify whether precision really scales by $2^{0.5}$, which would be a proof that
statistics limitation dominates over modeling uncertainty. Regarding factor (B), more
precise modeling developments with Monte Carlo (MC) simulations such as CORSIKA
and Geant4 may improve the accuracy. The current work is based on the assumption is
that the muons travel straight forwardly without experiencing scatterings; however, it
must be noted that after the muons are generated near the tropopause, they travel through
a material with a thickness equivalent to 20-meter water equivalent (m.w.e.) - 40 m.w.e.
for the muons arriving from an elevation angular region between 14 °- 32 °, having a
tendency to scatter to cause longer track lengths in the troposphere. This effect must be
taken into account in our future work to make improvements to factor (B). The
discrepancy between the balloon position and the muon generation region may also
influence how closely the compared data sets match. The muospheric layer thickness
seasonally oscillates, but its amplitude is likely to depend on location of the measurement
on the Earth since the near surface temperature is regionally varied. If the surface
temperature is different between the location underneath the ballon and the location
underneath the region of interest of the muopause, the muopause height in this region and
the balloon-based isobaric surface altitude will not coincide. Since the balloon's trajectory
is random, and it is difficult to control it, the next step in development is to compare the
DOF data with the satellite-based stratospheric sensing data.
**7. Conclusion**
In conclusion, a new muographic technique called DOF was proposed, and with this
technique, it was found that the muopause height interlocks with the isobaric surface
height in the upper troposphere and lower stratosphere. This work defined (A) the position
of the muopause and the layer span of the muosphere on the Earth; additionally, it was
shown that (B) the muopause is located in the lower stratosphere, (C) seasonal variations
in the muon flux are predominantly ruled by muopause dynamics, (D) muopause
dynamics can be visualized with DOF muography by taking advantage of directional

patterns of cosmic-ray muon's survival probabilities, and (E) muopause dynamics is closely related with isobaric surface height variations in the lower stratosphere.

Muopause dynamics has the potential to contribute to research focused on the upper tropospheric and lower stratospheric dynamics. In future studies, the potential of DOF muography for application to studying the dynamical processes occurring in the upper troposphere and lower stratosphere will be further investigated by performing related case studies and making specific comparisons with other atmospheric climate datasets. The next step would be spatiotemporal mapping of the muopause that would reflect spatiotemporal variations in tropospheric convection depth.

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

**Data Availability**

The datasets used and/or analyzed during the current study are available from the corresponding author on reasonable request.


**Author Contribution**

H.K.M.T. wrote the text. H.K.M.T. prepared the figures. H.K.M.T. reviewed the manuscript.


**Competing interests**

The author is a member of the editorial board of Geoscientific Instrumentation, Methods and Data Systems.




