# Peer review of "Distance of flight of cosmic-ray muons to study dynamics of the 1"

_Geoscientific Instrumentation, Methods and Data Systems, 2024_

## Referee Comment (RC1)

Dear,

Editorial support team
Copernicus Publications

Referee report of the manuscript gi-2024-4 (**Distance of flight of cosmic-ray muons to study dynamics of the upper muosphere**).

The paper presents the concept of muosphere as the region where a high concentration of muons and its limit respectively. It describes muosphere/muopause dynamics depending on crustal density (millennia time order) or isobaric surface height variations. The author introduces the DoF (Distance of Flight) technique for characterizing the muosphere based on measured data of the muon flux. The paper describes in detail the modeling process and compares it with measured data.

The technique is interesting but its application in a concrete problem isn't obvious. It's important to extend this technique in a real scenario and contrast it conceptually with techniques (like LIDAR) that can perform the same work. I recommend publishing the paper after adding the modifications listed below.

**line 49**. The muosphere is defined from $+10\,\mathrm{km}$ to $-10\,\mathrm{km}$, however in the equation at line 55 the muosphere thickness is $3.5{\times}10^4\,\mathrm{m}$.

**line 54**. Typo in $1.6 \times 10^2 m^{-2} s^{-1}$.

**line 53**. The estimated muon abundance is $\sim 8 \times 10^{12}$ muons, however solving the equation is $\sim 9.3 \times 10^{12}$ muons. I recommend to put the equation out from the text.

**line 53**. In the muon abundance equation a constant integral muon flux is used, this flux is the one measured at sea level, but not the same underground where the muon flux significantly decreases. Does it affect the abundance calculation?

**line 72**. The author establishes that study of muopause dynamics contribute to the research field and mention some cases. Could you describe, in more detail, how the parameterization of the muosphere/muopause can contribute?

**line 95**. Some reports on the study of barometric and temperature effects in the muon flux are mentioned. The Pierre Auger Observatory has made studies on the influence of atmospheric pressure/temperature (using meteorological balloons) in cosmic ray flux, it's important to cite them. J. Blumer et al. Atmospheric Profiles at the Southern Pierre Auger Observatory and their Relevance to Air Shower Measurement. 2005.

**line 102**. The author set three major identified characteristics, but only A and B are described, is there a C?

**Figure 2**. The scale of the muon flux variation is not clear in the panel A. Was equation 1 obtained form panel-B plot on Figure 2?

**Figure 3**. It's important to add error bars to the plots A and B due to they are based on data.

**Figure 4**. The number label on the left of the blue line is missing.

**Figure 5**. It's valuable to add the line legend to both plots. The plot description is split into two different pages and it makes the figure reading difficult.

**Figure 5**. Why does the curve behave different at $90°$ zenith? isn't expected to follow the same trend than for the other zenith angles?

**line 246**. If the zenith angle is close to zero the equation diverges, so "the spherical curvature of the Earth has to be considered (for $\theta = 90°$)". Does it mean that the angle dependence is taken into account only for $\theta = 90°$?

**Figure 6**. Add error bars to plots contained in Figure 6.

**line 270**. The zenith-angle integration range was set to be $50° - 80°$. Why?

**line 294**. Typo, a dot is missing. (. Three PSDs ...)

**line 295**. In the apparatus description, the author says that between detection layers two absorbing layers are set (10-cm thick lead block and a 3-cm thick stainless-steel), if you have 10 cm of lead, is it necessary to have 3 cm of stainless-steel? why?

**line 296**. Typo. "10-cm thick lead block and a 3-cm thick stainless-steel with a thickness of **3 cm**".

**line 240**. Write properly equation 2-2. It's confusing.

**Equation 2-2**. What is the meaning of the constant 660? where does it come from?

**Equation 1**. The relationship between atmospheric pressure and muon flux is inverse, in that way, isn't the slope of equation 1 negative?

---

## Referee Comment (RC2)

This is an interesting paper, healthily interdisciplinary, as it applies techniques from particle (astro)physics to the study of the atmosphere, leading to novel concepts. It is very well written.

Before accepting the paper, however, I recommend that some weaknesses of the data interpretation be addressed, as listed below. Moreover, I would have appreciated the author to elaborate more about the potential relevance of this study for atmospheric science.

My own research is mostly in particle physics, therefore I may be missing a few points that might be more obvious to the typical readers of this journal. However, I assume that this article should aim at being appreciated also outside of the core readership of this journal, therefore I suggest the author to assume that whatever is unclear to me might possibly deserve being elaborated more in the body of the paper.

**Main comments:**

The entire point of this paper is to extract knowledge about the atmospheric dynamics from measurements (performed at sea level) of observables that implicitly integrate through a very thick amount of atmosphere (from 30 to 0 km a.s.l.). Necessarily, this implies that in order to extract any interpretation from this method we need a reliable simulation of the atmospheric dynamic. This creates a sort of circular logic. Not being an atmospheric scientist, I would like to see the paper elaborate more explicitly on this point: how much can we rely on the models and which free parameters of the models can be pinned down with this new method?

I am not sure how to reconcile the assumptions of this paper with the similar study by Tramontini et al.:
https://agupubs.onlinelibrary.wiley.com/doi/10.1029/2019EA000655
In that paper, the muon flux is used to infer information about the middle-atmosphere dynamics such as SSW. I also note that SSW are mentioned at lines 75-76 of the present paper (but without citing Tramontini et al.).
My understanding is that the two studies are using the same probe to analyze the same medium, but under different assumptions and to infer different free parameters. If that's the case, what could be done to disentangle the middle- and upper- atmosphere dynamics in order to get less model-dependent results?
I suggest to include a citation to that paper (to which I am not related, therefore I might have misunderstood its message), accompanied by a critical discussion of similarity, difference and possible complementarity with the present study.

Similarly, I understand that this work makes also different assumptions with respect to the papers cited at lines 95-98. It is said at lines 99-100 that the difference is that those works focused on barometric and temperature effects. I suppose that the same can be said of the SSW study by Tramontini et al.? Anyway, a few more words would be appreciated here, to also address the previous point.

The paper separates the barometric effect and the isobaric height effect. Usually one has the barometric formula describing the functional dependence between the pressure and the height, in that case, the ground-level pressure should correlate with the height of the muon generation layer. May you please clarify whether by "barometric effect" it is meant here the effect of deviation from the canonical barometric formula?

Lines 151-152: not only the energy loss, but also the probability of meson-nucleus interactions. Can you argue that this has a negligible effect on your conclusions?

One of the crucial and most original developments in this paper is the proposal to use cyclone data for calibration. I find it clever, but again this makes several implicit assumptions: in essence, it is assumed that the barometric dependence of muon flux variations is universal. At the very least, I would like to see a proof that it is universal across different cyclones (and even that, probably, wouldn't be an air-tight proof that the calibration is valid in all contexts). Are there additional cyclone datasets that the author can analyse and compare, to 1) prove that the calibration curve extracted for one particular cyclone is valid for any cyclone, and 2) extract, from the variance of the fit parameters across all datasets, an additional systematic uncertainty to attach to this calibration. And it would be great if the author can find other data (not cyclone-related) where large pressure differences are observed, to be used to further validate the method.

Please perform a test of the sensitivity of the fitted formula of eq.1 to the time window during the Typhoon period, for example, do the coefficients change significantly if only considering the data points between 09/29 and 10/01 ?

How stable is the detector response? Implicit in all this analysis is that any observed variation in muon flux can only be due to factors that are external to the detector. But if e.g. the gain or the noise of the detector depend on some systematic effect, which is usually the case, this should be estimated. For the type of detector considered, for example, one can expect significant dependence on the **local** temperature around it. The statement at line 172 may be too optimistic. If the calibration procedure is such to correct for any temperature-dependent or time-dependent effect at the detector level, additional details must be provided about that.
All this is particularly relevant to support the statement at lines 322-323: one can imagine for example that a dependence of the detector response to the local temperature might have the negative correlation observed here, hence faking the effect sought to be studied. A possible test that shouldn't take too much effort could be a separation of the dataset between day and night.

Provide somewhere the goodness of fit test for the calibration curve of Fig. 2 (right).

It is mentioned at Lines 228-230 that the muon spectrum model is taken from three experimental papers. I think that more details are necessary here. Are these three measurements compatible between them? Were they combined (and if so, how)? Or are they covering disconnected ranges in momentum and zenith angle, such that there is no need of combining but there is a need for interpolating as explained in point B (lines 232-233)? Figure 5 only shows the interpolated model, I suggest to overlay the input data points on this plot (which would probably make my questions unnecessary).

The apparatus comprises slabs of both lead and steel. What is the purpose of having both? I assume that one is more effective against a type of background and the other against another type, but the current text is not explaining that, and it is probably not obvious.

As mentioned in Section 5 the muon flux is normalized to the value observed on Aug. 20, 2017. Are Delta H and Delta N in eq.(2-2) and Delta P and Delta N in eq.1 also the difference to the corresponding values of that day? If so, do the results change if a different reference day is chosen for everything?

In Figure 7 I suggest to overlay also the predicted flux without the barometric correction, to

quantitatively demonstrate that including the barometric correction indeed improves the compatibility between the model prediction and data.

Line 347 mentions limitations from (a) statistics and (b) modeling; it would be important to quantify their relative weight.
It is then mentioned that more statistics will be taken thanks to a larger detector, but at line 349 we are told that the size will be 4.5 m2, which is only twice the detector used for this paper. This means that the purely statistical uncertainty can only decrease by a factor of sqrt(2). Therefore, this is not going to change the game significantly: to reduce the statistical uncertainty by an order of magnitude, the detector area should increase by two orders of magnitude (the lateral size by an order of magnitude), hence it would be better to warn the reader about that. If the purpose of the 4.5 m2 detector is just to verify whether precision really scales by sqrt(2), which would be proof that statistics limitation dominates over modeling uncertainty, then better to state it explicitly.

Line 351 mentions ongoing studies with EAS Monte Carlos. Which MC programs are being used? CORSIKA?

**Editorial comments:**

Line 14: after "concentration of muons", in parenthesis, we see an approximate estimate of the amount of muons in the whole muosphere, but not a concentration. That would be muons/km^3. I suggest changing "concentration" into "number".

It is hinted at the end of the abstract that more case studies will have to be studied in the future, and the Conclusion sections ends with a similar statement. I find these statements a bit vague, I believe that it would be interesting to elaborate a bit, in the Conclusion, about which case studies may be interesting to investigate with this technique.

Lines 29-30 are an exact repetition of the beginning of the abstract.

Line 34: "usually do not"->"rarely"

Line 37: "start to"->"increasingly"; otherwise the reader unfamiliar with particle physics may think that this is an on/off type of process, instead of a continuous one.

Lines 39-40: "do not strongly interact": I suppose this is a reference to the Strong Nuclear Force, to which muons are insensitive, but this reference will not be obvious to the reader unfamiliar with particle physics. This may be solved by adding a few words to explain that muons only feel electromagnetic and weak nuclear interactions but not strong nuclear ones, or by modifying this sentence to be more specific.

Line 46: "due to its"->"due to their" (the subject is muons, plural)

Line 50 defines the muosphere between +30 and -10 km from sea level but at line 55 the calculation uses a thickness of 35 km, instead of 40.

Line 64: repetition ("and and").

Line 72: "surface height variations. Studying"->"surface height variations, studying"

Line 75: first occurrence of SSW acronym, must be explained.

Line 85: first occurrence of DoF acronym, must be explained.

Line 88: "decay constant"->"mean lifetime at rest". In fact, the former is defined as the inverse of the latter. By the way, perhaps not all readers of this journal are familiar with relativistic time dilation, so I would advice to spend a few words to say that while the lifetime at rest is 2.2 microseconds, the actual lifetime as observed from the detector, and therefore the path traveled, is much longer by the relativistic time dilation factor, and that the latter depends on energy. You may also give typical ranges, to be even more informative to non-experts.

Line 98: "recent detailed review": being from 2011, it is not so recent anymore.

Line 114: "hadronic process": be more specific, e.g. "hadronic collisions of mesons with nuclei"; "decay process": add "of the mesons" or rephrase somehow.

Line 127: "detail"->"detailed"

Line 166: Delta P with respect to which reference P value?

Line 174: remove "which".

Line 196: add "estimated" before "variations".

Line 197: explain that B is based on Equation 1.

Line 202: I suggest to expand the title: studies of what?

Line 204: after "to monitor" and after "measurement results", explain which observables are being monitored / measured. Only much later in the section the reader is finally informed about that.

Line 216: I am not sure if "geopotential altitude" is a universally known term, I suggest giving its definition in parenthesis or in a footnote.

Line 269: racker → tracker

Line 281: remove the word "Reference"

Lines 295-296: I would remove the parenthesis around "such as positrons/electrons"

Line 296: "lead block"->"lead slab" (I presume), and also add "slab" after "stainless-steel"

Lines 296-297: remove "with a thickness of 3 cm", as that information already appeared few words earlier.

Line 330: I suggest to express this number in percent, i.e. 0.987% (as you do at line 342 for its approximation).

Lines 375 and 380: I suppose that LOF here should be replaced by DoF?

Line 453: no better reference? According to Google Scholar, this might be arXiv:1909.01406 and PoS ICRC2019 (2020) 894.
Note by the way that clicking on the link redirects to [https://0.0.1.197/](https://0.0.1.197/)

Lines 458-459: I suppose, reading the arXiv entry, that these proceedings have been then published somewhere. Please update the reference.

---

## Author Comment (AC1)

RC1

(1) The paper presents the concept of muosphere as the region where a high concentration of muons and its limit respectively. It describes muosphere/muopause dynamics depending on crustal density (millennia time order) or isobaric surface height variations. The author introduces the DoF (Distance of Flight) technique for characterizing the muosphere based on measured data of the muon flux. The paper describes in detail the modeling process and compares it with measured data. The technique is interesting but its application in a concrete problem isn't obvious. It's important to extend this technique in a real scenario and contrast it conceptually with techniques (like LIDAR) that can perform the same work.

[AUTHOR]:

UV and visible light used for LiDAR are scattered/reflected by cloud and thus, ground-based LiDAR cannot be used for measuring the tropopause height when thick cloud covers the sky while the satellite-based LiDAR is not influenced by such lower cloud. However, researchers seem to utilize LiDAR to detect the reflection of the cloud located near the tropopause to indirectly acquire its height. If there is no cloud, satellite-based LiDAR technique cannot be applied.

https://www.researchgate.net/publication/238780374_Study_of_Tropopause_Height_over_Bueno s_Aires_Monitored_with_lidar_system

Muons are generated most extensively at the location where the density gradient accidentally matches with the density gradient near the tropopause. If the tropopause shifts upward, this density gradient shifts upward accordingly, and accordingly, the muopause shifts upward, then the muon's travel distance increases. The current technique measures the height of where muons are extensively generated; hence the height of the tropopause by measuring the muon's distance of flight (DOF). Due to the muon's strong penetrating capability, the DOF technique is not influenced by the cloud existence.

**Editorial comments will be applied to the draft in the next step. It seems this journal's system requires sending the response first and then subsequently revising the draft.**

=====

RC2

(1) The entire point of this paper is to extract knowledge about the atmospheric dynamics from measurements (performed at sea level) of observables that implicitly integrate through a very thick amount of atmosphere (from 30 to 0 km a.s.l.). Necessarily, this implies that in order to extract any

interpretation from this method we need a reliable simulation of the atmospheric dynamic. This creates a sort of circular logic. Not being an atmospheric scientist, I would like to see the paper elaborate more explicitly on this point: how much can we rely on the models and which free parameters of the models can be pinned down with this new method?

[AUTHOR]:
The entire point of this paper is NOT from measurements of observables that implicitly integrate through a very thick amount of atmosphere.

The entire point of this paper is to measure the geometrical distance the location where the muons are generated (muopause) and the location where the muons are detected.

The "geometrical distance" is much clearer physical quantity than "areal density integrated over a very thick amount of atmosphere" that includes various unknown parameters.

For most of the cosmic muons (< 50GeV) the muon flux observed at sea level is compilation of (A) the barometric effect and (B) the muopause height variation effect. Factor (A) has been well studied in the prior work:
https://www.nature.com/articles/s41598-022-20039-4
The current study clarified that factor (B) is larger than factor (A), and factor (B) can be well estimated by applying JMA barometric correction.

(2) I am not sure how to reconcile the assumptions of this paper with the similar study by Tramontini et al.:
https://agupubs.onlinelibrary.wiley.com/doi/10.1029/2019EA000655
In that paper, the muon flux is used to infer information about the middle-atmosphere dynamics such as SSW. I also note that SSW are mentioned at lines 75-76 of the present paper (but without citing Tramontini et al.).
My understanding is that the two studies are using the same probe to analyze the same medium, but under different assumptions and to infer different free parameters. If that's the case, what could be done to disentangle the middle- and upper- atmosphere dynamics in order to get less model-dependent results?
I suggest to include a citation to that paper (to which I am not related, therefore I might have misunderstood its message), accompanied by a critical discussion of similarity, difference and possible complementarity with the present study.

[AUTHOR]:

Thanks for letting me know this interesting work done in 2019 which I embarrassingly didn't know. I will include this reference to this work.

As is written in this paper:

"An increase in the atmospheric temperature lowers the atmospheric density. Temperature changes in the atmosphere may therefore affect the production of muons (Gaisser et al., 2016), " These authors applied the muon's barometric effect to their atmospheric study.

And what they measured was "observables that implicitly integrate through a very thick amount of atmosphere."

There are three phenomena that affects the muon flux (A) variations in hadronic interaction MFPs due to stratospheric temperature variations, (B) variations in the muopause height due to variations in depth of tropospheric convection; hence variations in the muon's distance of flight between the muopause and sea level, and (C) variations in the muon's range due to barometric variations. There are three effects in total. Among these, factor (A) is prominent only for high energy muons (> 50 GeV). Tramontini et al. applied factor (C) to their work to study atmospheric density variations. On the contrary, factor (B) was applied to this work. This will be clarified in the draft.

(3) Similarly, I understand that this work makes also different assumptions with respect to the papers cited at lines 95-98. It is said at lines 99-100 that the difference is that those works focused on barometric and temperature effects. I suppose that the same can be said of the SSW study by Tramontini et al.? Anyway, a few more words would be appreciated here, to also address the previous point.

[AUTHOR]:

The aforementioned issues will be more clarified in the draft.

(4) The paper separates the barometric effect and the isobaric height effect. Usually one has the barometric formula describing the functional dependence between the pressure and the height, in that case, the ground-level pressure should correlate with the height of the muon generation layer. May you please clarify whether by "barometric effect" it is meant here the effect of deviation from the canonical barometric formula?

[AUTHOR]:

Yes, your understanding is correct. The ground-level pressure correlates with the muopause height. If the ground surface is heated, more convection grows, the pressure decreases, and the muopause

height increases. The muon flux we observe at sea level is a compilation of the factors (B) and (c).

However, factor (B) can be separated from factor (C) because these factors influence to physically independent processes (energy loss process & decay process); hence these processes can be independently quantified, and can be merged at the end. This will be clarified in the draft.

(5) Lines 151-152: not only the energy loss, but also the probability of meson-nucleus interactions. Can you argue that this has a negligible effect on your conclusions?

[AUTHOR]:
The influence of variations in hadronic interaction MFP to the muon flux is only limited to the muon's energy range only above 50 GeV [1] (for pions to travel a sufficient distance to collide with another nuclei), and it is not the energy range we discuss here.
[1] The IceCube Collaboration. Seasonal variation of atmospheric muons in IceCube (2019). Retrieved from https://par.nsf.gov/servlets/purl/10171530

The fraction of the muons above 50 GeV to the entire cosmic muons is less than 0.5% and thus, this hadronic effect is negligible.

(6) One of the crucial and most original developments in this paper is the proposal to use cyclone data for calibration. I find it clever, but again this makes several implicit assumptions: in essence, it is assumed that the barometric dependence of muon flux variations is universal. At the very least, I would like to see a proof that it is universal across different cyclones (and even that, probably, wouldn't be an air-tight proof that the calibration is valid in all contexts). Are there additional cyclone datasets that the author can analyse and compare, ...

[AUTHOR]:
One of the crucial and most original developments in this paper is NOT the proposal to use cyclone data for calibration. The author doesn't think "the use of cyclones for the calibration of the method" is so original and main contribution to this paper. This has been already established at https://www.nature.com/articles/s41598-022-20039-4.
The relationship between the muon flux and the pressure drop by the cyclone passage is well documented in this reference (see Figure 1, Figures 6-8.)
Figure 2 in the current work simply shows the consistency with the prior work: 1.5%/10hPa.

The crucial and most original developments in this paper is the "tropopause scanner (somewhat like

atmospheric LiDAR (ATLID))" by measuring the distance of flight of cosmic muons.

By applying this technique, the tropopause height distribution can be map out for studying the atmospheric radiative balance. As far as I know, this methodology has never been appeared in the history of cosmic ray science.

(7) How stable is the detector response? Implicit in all this analysis is that any observed variation in muon flux can only be due to factors that are external to the detector. But if e.g. the gain or the noise of the detector depend on some systematic effect, which is usually the case, this should be estimated. For the type of detector considered, for example, one can expect significant dependence on the local temperature around it. The statement at line 172 may be too optimistic. If the calibration procedure is such to correct for any temperature-dependent or time-dependent effect at the detector level, additional details must be provided about that.

All this is particularly relevant to support the statement at lines 322-323: one can imagine for example that a dependence of the detector response to the local temperature might have the negative correlation observed here, hence faking the effect sought to be studied. A possible test that shouldn't take too much effort could be a separation of the dataset between day and night.

[AUTHOR]:

The detectors are the same model that have been used for

https://www.nature.com/articles/s41598-021-98559-8

The results shown in Figures 6-8 indicate that the detectors have stably operated for many years.

However, it is difficult to quantify the detector's stability from Figures 6-8 because barometric information is included in there.

The best way to check the long-term detector stability is to use IBE (inverse barometric effect = barometric variations are compensated by the tidal height variations = total areal density above the detector will be constant as long as the detector is located undersea), so that the local fluctuations due to barometric variations can be intrinsically removed from the data without any artificial actions (such as subtracting a modeled value).

I prepared one figure below to show how barometric variations are well canceled out by tidal height variations in Tokyo bay. Therefore, if muons are measured underneath Tokyo bay, the muon flux there is not influenced by barometric variations. According to this, residual barometric fluctuation is suppressed within a few hPa at the maximum.

[Figure]

Figure. Example of the temporal variations in the tide anomaly observed at the Chiba Tide Gauge Station, Japan

As the next step, I show you the stability of the detector that consists of R7724 and EJ200 (the same configuration used in this work). The data were taken at the underwater tunnel called the Trance Tokyo-bay Aqua Line which is the busiest highway tunnel in Japan. The results are shown in Figure 4 at

https://www.nature.com/articles/s41598-021-98559-8.

The muon rate is almost constant and the average muon counts per lunar day (to cancel out the astronomical tide) and standard deviation (S.D.) were respectively 1,144,288 and 3187 (~ 2.8 per mille including statical errors of ~1 per mille) for more than 5-month measurements. The temperature inside the tunnel varies more than 20 degrees between winter and summer because it is a tunnel that opens to the ground surface.

(The tendency of decrease in muon counts comes from the thermal expansion of the pacific ocean.)

(8) Provide somewhere the goodness of fit test for the calibration curve of Fig. 2 (right).

[AUTHOR]:
Will be provided.

(9) It is mentioned at Lines 228-230 that the muon spectrum model is taken from three experimental papers. I think that more details are necessary here. Are these three measurements compatible between them? Were they combined (and if so, how)? Or are they covering disconnected ranges in

momentum and zenith angle, such that there is no need of combining but there is a need for interpolating as explained in point B (lines 232- 233)? Figure 5 only shows the interpolated model, I suggest to overlay the input data points on this plot (which would probably make my questions unnecessary).

[AUTHOR]:
The data points will be overlayed.

(10) The apparatus comprises slabs of both lead and steel. What is the purpose of having both? I assume that one is more effective against a type of background and the other against another type, but the current text is not explaining that, and it is probably not obvious.

[AUTHOR]:
A stainless-steel case is needed to protect the lead for the following reasons.
(A) Lead is soft, and for long-time measurements, it is deformed, so it has to be supported by the stainless-steel case.
(B) Lead is poisonous. Not allowed to use it outdoor environments without coverage, so it has to be covered by the stainless-steel box.

(11) As mentioned in Section 5 the muon flux is normalized to the value observed on Aug. 20, 2017. Are Delta H and Delta N in eq.(2-2) and Delta P and Delta N in eq.1 also the difference to the corresponding values of that day? If so, do the results change if a different reference day is chosen for everything?

[AUTHOR]:
Firstly, Figure 7 shows a relative muon flux: how it is varied with respect to the flux observed on Aug. 20, 2017.
Secondly, we are discussing to derive relative height variations (Delta H) from relative differential flux variations ($I_2/I_1$)
For example,

On Aug. 20, 2017,
$I_1 = I_0 \exp(-\text{Delta } H_1 (\sin \theta)^{-1}/660 \gamma (E)[m])(1-\text{Delta } N_1)$
$n_1$= corresponding integrated flux
On Oct. 20, 2017
$I_2 = I_0 \exp(-\text{Delta } H_2 (\sin \theta)^{-1}/660 \gamma (E)[m])(1-\text{Delta } N_2)$

$n_2$= corresponding integrated flux

$(n_1)/(n_2)=$
¥int[exp(-Delta $H_1$(sin theta)$^{-1}$/660 gamma (E)[m])(1-Delta $N_1$)]/
¥int[exp(-Delta $H_2$ (sin theta)$^{-1}$/660 gamma (E)[m])(1-Delta $N_2$)]
Since Delta $N_1$/Delta $N_2$ can be derived from Eq. (1)
Delta $H_1$ - Delta $H_2$ is given.
This technique is to derive variations of the tropopause height, doesn't derive the absolute tropopause height.

In order to derive the absolute height $H_2$, additional measurements are needed e.g. balloon experiment. If $H_1$ is given from other measurements, $H_2$ can be derived.

(12)
In Figure 7 I suggest to overlay also the predicted flux without the barometric correction, to quantitatively demonstrate that including the barometric correction indeed improves the compatibility between the model prediction and data.

Will be overlayed.

(13)
Line 347 mentions limitations from (a) statistics and (b) modeling; it would be important to quantify their relative weight.
It is then mentioned that more statistics will be taken thanks to a larger detector, but at line 349 we are told that the size will be 4.5 m2, which is only twice the detector used for this paper. This means that the purely statistical uncertainty can only decrease by a factor of sqrt(2). Therefore, this is not going to change the game significantly: to reduce the statistical uncertainty by an order of magnitude, the detector area should increase by two orders of magnitude (the lateral size by an order of magnitude), hence it would be better to warn the reader about that. If the purpose of the 4.5 m2 detector is just to verify whether precision really scales by sqrt(2), which would be proof that statistics limitation dominates over modeling uncertainty, then better to state it explicitly.

[AUTHOR]:
Currently, modeling and statistics (~1% SD) are comparable. As can be seen in Figure 7, there is a systematic discrepancy (1-2%) between the modeling results and the actual flux Dec 2017-Aug 2018, Oct 2018-Jan 2019, Jul 2019-Sep 2019. The statistics can be improved by increasing the size of the

detector (as you say) by two orders of magnitude. But practically, it is difficult to increase the lateral size by an order of magnitude, due to the space limitation and the costs. So, as the first step, the detector will be increased to a double size (4.5 m2) to verify whether precision really scales by sqrt(2), which would be proof that statistics limitation dominates over modeling uncertainty.

This will be clarified in the draft.

(14) Line 351 mentions ongoing studies with EAS Monte Carlos. Which MC programs are being used? CORSIKA?

[AUTHOR]
This statement is incorrect since developments are not ongoing.
This phrase will be revised to

...more precise modeling developments with Monte Carlo (MC) simulations such as CORSIKA and Geant4 may improve the accuracy.

**Editorial comments will be applied to the draft in the next step. It seems this journal's system requires sending the response first and then subsequently revising the draft.**

======
RC3
(1) The paper proposes an original link between sea-level muon flux measurements and assessment on the atmosphere state's equation in a whole. It introduces a neologism "muosphere" defined as the part of the atmosphere + geosphere where muon flux is relevant, in a way which still not completely clear after the reading of the article, especially on the lower border (the geosphere one) since muons penetrate the Earth over distances depending on their incident energy. The article focuses on the "muopause" but I do not see much considerations on the underground probing of the geosphere. Please develop more on both borders of your "muosphere".

[AUTHOR]
The paper's main focus is not underground probing of the geosphere. Discussing about the lower border in the geosphere is irrelevant from the DoF technique which is the focus of this paper.

However, I agree it is better to complete the description by adding the the lower border in the geosphere.

(2) The main weakness of this article is that, apart from the invention of the term "muosphere", it does not propose innovative approaches on the use of muons fluxes o derive properties of the atmosphere. It has been published in the past that muons flux is correlated to barometric observables, stratospheric temperatures etc and be used to detect transient phenomena of abrupt changes in those parameters (e.g. SSW as mentionned but not cited in the article) or to anticipate violent phenomena such as storms.

[AUTHOR]

See the replies to (1) of RC1 and (1) and (2) of RC2.

The atmospherics dynamic processes in the upper troposphere (UT) and the lower stratosphere (LS) have much interest due to their important role and impact on the atmospheric radiative balance. Factor (A) and factor (C) do not provide information to this issue, and only factor (B) has a potential to address this issue. This is the main topic of this paper.

(3) TOF methods are also well documented and I would not focus the title of the article on this item.

[AUTHOR]

Firstly, the author doesn't know where TOF methods are well documented. If you could provide some examples, that will be highly appreciated.

Secondly, the currently discussing topic is not based on the "muon's time of flight (TOF)". We don't measure the time muons travel from the point at which the muons are generated to the ground surface. What we measure is variations in survival rate of muons to calculate variations in muon decay rate; hence variations in the distance they traveled.

As a response to upper atmospheric dynamics, if the muopause height ascends and descends, the averaged travel lengths of muons are respectively extended and contracted. Muons tend to decay more if the muopause height ascends, and muons tend to decay less if the muopause height descends, consequently, the sea level muon flux decreases and increases as a response to the ascent and descent of the muopause. This effect has been known since many decades ago, however, as far as I know there has been no attempt to inversely use this effect to measure the distance between the muopause and the ground surface.

(4) The real originality of the paper is the use of cyclones for the calibration of the method. I would

suggest to focus more on that point and try to implement a concept close to the one of "standard candles" used is cosmology with the supernovae for instance. Please try to elaborate on this item and provide more details on the analysis tools, goodness-of-fit, likelihood analysis to assess whether the results of this method may be reproducible.

[AITHOR]
See the reply to (6) of RC2.

CC
See the replies to (1) of RC1 and (1) and (2) of RC2.

(1) Despite the meticulous writing, there are some numerical errors:

[AUTHOR]
Will be corrected.

(2) I would ask the author to be more specific in explaining how the values between lines 172 and 179 were calculated.

[AUTHOR]
60%, 82%, and 100% for deviations of ≤1s, ≤1.5s, and ≤2s can be derived just by counting the number of data points. I imagine SD is self-explanatory.